# Fungal Species and Mycotoxins Associated with Maize Ear Rots Collected from the Eastern Cape in South Africa

**DOI:** 10.3390/toxins16020095

**Published:** 2024-02-08

**Authors:** Jenna-Lee Price, Cobus Meyer Visagie, Hannalien Meyer, Neriman Yilmaz

**Affiliations:** 1Department of Biochemistry, Genetics and Microbiology, Forestry and Agricultural Biotechnology Institute (FABI), University of Pretoria, Pretoria 0028, South Africa; jenna-lee.price@fabi.up.ac.za (J.-L.P.); cobus.visagie@fabi.up.ac.za (C.M.V.); 2Southern African Grain Laboratory (SAGL), Grain Building-Agri Hub Office Park, 477 Witherite Street, The Willows, Pretoria 0040, South Africa; hannalien.meyer@sagl.co.za

**Keywords:** maize, *Fusarium* spp., LC-MS/MS, multi-mycotoxins, ear rot

## Abstract

Maize production in South Africa is concentrated in its central provinces. The Eastern Cape contributes less than 1% of total production, but is steadily increasing its production and has been identified as a priority region for future growth. In this study, we surveyed ear rots at maize farms in the Eastern Cape, and mycotoxins were determined to be present in collected samples. Fungal isolations were made from mouldy ears and species identified using morphology and DNA sequences. Cladosporium, Diplodia, Fusarium and Gibberella ear rots were observed during field work, and of these, we collected 78 samples and isolated 83 fungal strains. *Fusarium* was identified from Fusarium ear rot (FER) and Gibberella ear rot (GER) and *Stenocarpella* from Diplodia ear rot (DER) samples, respectively. Using LC-MS/MS multi-mycotoxin analysis, it was revealed that 83% of the collected samples contained mycotoxins, and 17% contained no mycotoxins. Fifty percent of samples contained multiple mycotoxins (deoxynivalenol, 15-acetyl-deoxynivalenol, diplodiatoxin and zearalenone) and 33% contained a single mycotoxin. *Fusarium verticillioides* was not isolated and fumonisins not detected during this survey. This study revealed that ear rots in the Eastern Cape are caused by a wide range of species that may produce various mycotoxins.

## 1. Introduction

Maize is a staple crop produced by commercial, small-scale and subsistence farmers in many African countries, especially in sub-Saharan Africa [1]. In South Africa, commercial maize production is concentrated in the central regions that includes the Free State, North West, Gauteng and Mpumalanga provinces [2]. The Eastern Cape (EC) contributes about 1% of South Africa’s annual maize production, with many farms classified as small-scale or emerging across the province [3].

Maize production in Africa continuously faces challenges like pests and diseases preventing regions from reaching their full production potential [4]. Maize ears are prone to fungal infection, often resulting in ear rots and/or subsequent mycotoxin contamination of kernels. Economically, it is one of the most important diseases [5], either directly associated with reduced trade revenue due to contamination, or with adverse health effects [1].

Globally the most common maize fungal ear rots include Aspergillus (AER), Cladosporium (CER), Diplodia (DER), Fusarium (FER), Gibberella (GER) and Penicillium (PER) [6]. The distribution of ear rots is dependent on climatic and geographical conditions; however, DER, FER and GER are reported as the most common to South Africa [5,7,8]. Diplodia ear rot is caused by *Stenocarpella maydis* (=*Diplodia maydis*)*,* whereas FER is caused by several species classified within the *Fusarium fujikuroi* species complex (FFSC) and GER by members of the *Fusarium sambucinum* species complex (FSAMSC) [9,10,11,12]. *Fusarium verticillioides* (=*F. moniliforme*) is most commonly linked with FER, while *F. graminearum* is most commonly linked with GER [13,14,15]. However, several other species from both the FFSC and FSAMSC can also cause ear rots and differ in their abilities to produce mycotoxins [13,16]. These mycotoxins can cause disease or death in animals and humans if consumed above threshold levels [17]. Mycotoxin contamination of maize can occur across the entire value chain, from the field, during transportation or storage, and is affected by many biotic and abiotic factors (e.g., climatic conditions) [18,19,20,21]. Fumonisins were first discovered from home-grown maize in the EC after being linked to oesophageal cancer in the Transkei region [22,23,24]. Since the discovery of fumonisins, mycotoxin levels in commercially produced maize in South Africa are monitored annually by the Southern African Grain Laboratory (SAGL) as part of the maize quality crop survey, but this does not usually include EC-produced maize.

The occurrence of maize ear rots in South Africa were previously reported [7,23,25,26,27,28,29,30]. In these studies, fungal species were identified using morphology or species-specific quantitative PCR (qPCR). However, the use of morphology often results in misidentifications, especially in the case of cryptic species (species that are morphologically identical but genetically different), while identification using species-specific qPCR does not consider the possibility of the presence of other potential ear rot causal agents. Furthermore, even though mycotoxin contamination of maize in South Africa has been studied [31,32,33,34,35], the correlation between ear rots and species associated with it has not been linked, especially using modern taxonomic approaches.

To help mitigate potential mycotoxin contamination in EC-produced maize, it is important to complete fungal surveys to determine what species are present on maize, which mycotoxins they may produce and determine the level of mycotoxin contamination across the province. This study thus aimed to complete a maize ear rot survey across EC farms, with the goals to (1) determine their distribution and severity, (2) identify the fungal species responsible for ear rots and (3) determine the mycotoxins present on collected ear rot samples.

## 2. Results

### 2.1. Ear Rot Survey

Ear rots were observed at 12 of the 19 farms (Table 1). These include CER (five farms, Figure 1A), DER (six farms, Figure 1B), FER (ten farms; Figure 1C), and GER (three farms; Figure 1D). Generally, ear rot incidence (0.8–14%) and severity (an average score of two recorded) was low (Table 1). The exception was farm 5, where the field had mild-to-severe DER, attributed to a hailstorm that damaged ears a month earlier. From the farms where ears rots were reported, a single ear rot was observed at four farms including farm 3 (FER), farm 8 (CER), farm 9 (CER) and farm 11 (FER). Farms where two types of ear rots were observed include farm 2 (DER and FER), farm 4 (CER and FER), farm 6 (DER and FER), farm 7 (FER and GER) and farm 10 (DER and FER). Three types of ear rots were observed at farm 1 (CER, DER and FER) and farm 5 (DER, FER and GER), while four types of ear rots were observed at farm 12 (CER, DER, FER and GER).

### 2.2. Mycotoxins

Aflatoxins (AFs; including AFB_1,_ AFB_2,_ AFG_1_ and AFG_2_), fumonisins (FBs; including FB_1,_ FB_2_ and FB_3_), HT-2 toxin (HT-2), ochratoxin A (OTA) and T-2 toxin (T-2) were not detected at any farm. However, deoxynivalenol (DON), 15-acetyldeoxynivalenol (15-ADON), diplodiatoxin (DIP) and zearalenone (ZEN) were detected (summarized in Table 2). Multiple mycotoxins were detected from six of the twelve farms, including farm 2 (DON, 15-ADON and DIP), farm 4 (DON and 15-ADON), farm 5 (DON, 15-ADON, DIP and ZEN), farm 6 (DON and DIP), farm 7 (DON, 15-ADON and ZEN) and farm 12 (DON, 15-ADON, DIP and ZEN). A single mycotoxin was detected at farm 1 (DIP), farm 8 (DON), farm 9 (DIP) and farm 10 (DIP), while none of the tested mycotoxins were detected from farms 3 and 11, even though FER was detected at both.

Deoxynivalenol was detected from seven farms. It was detected from four farms where GER samples were collected, including farm 5 (5678 µg/kg), farm 6 (254 µg/kg), farm 7 (18,306 µg/kg) and farm 12 (7273 µg/kg). It was also detected from three farms (farm 2 (879 µg/kg), farm 4 (3143 µg/kg) and farm 8 (89 µg/kg (<LOQ)) where GER was not observed and no FSAMSC species were isolated.

The DON derivative, 15-ADON, was detected from all farms where DON was detected, except for farms 6 and 8. It was detected from three farms where GER samples were collected including farm 5 (257 µg/kg), farm 7 (1263 µg/kg) and farm 12 (736 µg/kg). It was also detected from farms where GER was not observed and no FSAMSC species were isolated, including farm 2 (174 µg/kg) and farm 4 (301 µg/kg). The levels of 15-ADON detected were far lower than DON.

Diplodiatoxin was detected from six farms where DER was collected, including farm 1 (726 µg/kg), farm 2 (122 µg/kg), farm 5 (33,808 µg/kg), farm 6 (2648 µg/kg), farm 10 (1268 µg/kg) and farm 12 (23,004 µg/kg), but also from farm 9 (25 µg/kg (<LOQ)) where no DER or producing species was observed. Lastly, ZEN was detected from three farms (farm 5 (23 µg/kg), farm 7 (639 µg/kg) and farm 12 (67 µg/kg)) where GER was collected.

### 2.3. Fungal Identification

Isolations from 78 ear rot samples resulted in 83 fungal strains identified into six genera and 16 species, including *Cladosporium* (two species), *Epicoccum* (one species), *Fusarium* (nine species), *Mucor* (one species), *Stenocarpella* (two species) and *Trichoderma* (one species). The *Epicoccum* (n = 1), *Mucor* (n = 1) and *Trichoderma* (n = 1) were not associated with any specific ear rot symptoms. Strains isolated, species identified and gene sequences generated with GenBank accession numbers are summarized in Appendix A. From FER samples, six *Fusarium* species (Figure 2A) were isolated and identified, including *F. clavus* (n = 1, *Fusarium incarnatum-equiseti* species complex (FIESC)), *F. oxysporum* (n = 2, *Fusarium oxysporum* species complex (FOSC)), *F. scirpi* (n = 1, FIESC), *F. sporodochiale* (n = 1; *Fusarium chlamydosporum* species complex (FCSC)), *F. subglutinans* (n = 4, FFSC) and *F. temperatum* (n = 33, FFSC). *Fusarium temperatum* was isolated from all farms where FER was observed, and this was the only species isolated from FER on farm 3. Three *Fusarium* species were identified from GER, and all belong to the FSAMSC (Figure 2B). These include *F. boothii* (n = 7), *F. graminearum* (n = 4) and *F. poae* (n = 3).

*Stenocarpella maydis* (n = 17) and *S. macrospora* (n = 1) were isolated from DER samples (Figure 2C). *Stenocarpella maydis* was isolated from all farms where DER was identified, with a single isolate of *S. macrospora* isolated from a DER sample together with *S. maydis*. *Cladosporium cladosporioides* (n = 4) and *C. pseudocladosporioides* (n = 2) were isolated from CER (Figure 2D).

## 3. Discussion

This is the first survey conducted in the EC of South Africa to make use of DNA sequences to accurately identify species associated with maize ear rots. Ear rots were observed at 12 of 19 farms surveyed, including CER (5/12), DER (6/12), FER (10/12) and GER (3/12). Fungal isolations from these resulted in the identification of six genera and 16 species, most notably *Fusarium* and *Stenocarpella*. Mycotoxins (DON, 15-ADON, DIP and ZEN) were detected at 10 of the 12 farms where ear rots were collected.

Fusarium ear rot has been reported frequently in South Africa [8,25,26,29,36] and worldwide [13,37,38,39,40,41,42,43], with *F. verticillioides* being the most common species reported, followed by *F. subglutinans* and *F. temperatum*. During this study, no *F. verticillioides* or FBs were detected from FER samples. This differs from previous South African surveys that report *F. verticillioides* as the most common species associated with FER [8,25,26,29,36]. The absence of *F. verticillioides* and FBs was surprising considering *F. verticillioides* and its mycotoxin, FB_1_, have been frequently reported from smallholder-produced maize in the EC [23,44,45,46]. During this study, *F. subglutinans* and *F. temperatum* were frequently isolated from FER. These are cryptic sister species that are reproductively isolated and were previously reported as distinct groups (1 and 2) of *F. subglutinans* [47,48]. The two groups were later described as two distinct species, *F. subglutinans* (previously Group 2) and *F. temperatum* (previously Group 1) [49]. *Fusarium subglutinans* is prevalent in cooler maize production areas such as the northern United States of America and Canada [50,51] and central and northern Europe [14]. Recently, *F. temperatum* has been found to dominate maize production areas in North America, Argentina and Europe [12]. Previous studies conducted on maize in South Africa detected *F. subglutinans* [25,26,29,36]. However, these studies were conducted prior to the reclassification and splitting of *F. subglutinans* into two species. This implies that species identifications in the South African literature may be outdated. It was previously reported that both *F. subglutinans* and *F. temperatum* can produce FBs [11,49,52]. However, the biosynthetic gene clusters required for FB production have not been found in the former species [38,53]. Whilst these species are sister species, their mycotoxin production potential differs, and they are known to produce emerging mycotoxins [11]. *Fusarium subglutinans* can produce moniliformin (MON), which is cytotoxic and has been implicated in protein synthesis inhibition, chromosome damage, intestinal haemorrhaging, coma and death in poultry [12,38,54], while *F. temperatum* can produce beauvercin (BEA), which has been implicated in the induction of apoptosis and enzyme inhibition [11,49,53,55]. Neither of these were tested for during our mycotoxin analyses.

Gibberella ear rot was previously reported in South Africa [8,25,26,29,56] and globally [13,40,41,43], with *F. graminearum* the most commonly reported. This survey identified known GER causal agents *F. boothii*, *F. graminearum* and *F. poae* [57]. Gibberella ear rot is found to predominate in cooler temperatures and regions with high precipitation [13,14,50,51]. Previous surveys from South Africa mostly reported *F. graminearum* as the GER causal agent [25,26,36]; however, *F. boothii* has also been reported [56]. *Fusarium poae* is a pathogen typically associated with small grain cereals and causes Fusarium Head Blight (FHB) of wheat in South Africa [56,58,59]. It has previously been isolated from asymptomatic maize in South Africa [46,60]; however, this is the first report of it being isolated from GER in the country. *Fusarium boothii* and *F. graminearum* are producers of DON (a regulated mycotoxin, <2000 µg/kg for human consumption) and its derivative 15-ADON (an unregulated mycotoxin) [61,62]. In addition to these mycotoxins, *F. graminearum* produces ZEN (a regulated mycotoxin, <75 µg/kg for human consumption) [63], while *F. poae* can produce emerging mycotoxins like diacetoxyscirpenol (DAS), fusarenone X (FUS-X), neosolaniol (NEO) and nivalenol (NIV) [11]. Deoxynivalenol, 15-ADON and ZEN were detected in this study and have also been detected from other maize-producing regions in South Africa [31,32,33,34,64,65,66]. Mycotoxin analysis revealed that the EC has one of three DON chemotypes (15-ADON chemotype), and this has previously been reported in South Africa, as well as the Americas and Europe [11,67].

In South Africa, annual mycotoxin surveys on various agricultural produce, including maize, are conducted. A shift in the dominant mycotoxins detected from all agricultural produce has been observed. Previously, reports indicated FB_1_ to be the dominant mycotoxin, especially associated with maize [23,32,34,35,65,68,69]. This study, and the above-mentioned studies have observed that FB_1_ is no longer the dominant mycotoxin and that DON, 15-ADON and ZEN are becoming more prominent. Health implications following consumption of DON include feed refusal by swine and altered immune function, while consumption of ZEN results in oestrogenic activity of mammals [13,50,62,70].

Like FER and GER, DER has been reported in South Africa [25,26,29,30,36], with *S. maydis* the main causal agent, but *S. macrospora* has also been shown to be associated with DER [26,71,72]. Diplodia ear rot is often found in maize production regions that experience dry weather early in the season, followed by wet weather conditions prior to and after silking conditions [71,73]. Outbreaks of DER typically occur in the late winter months [36]. This study isolated *S. maydis* from all DER samples, with a single isolate of *S. macrospora* found together with the former. *Stenocarpella maydis* is known to produce DIP and chaetoglobosins K and L; however, the production potential has been found to differ between strains [72,74,75]. Diplodiatoxin is an unregulated neuromycotoxin affecting mainly cattle and sheep and has been linked to diplodiosis outbreaks in South Africa, Argentina and Brazil [74,76,77]. Symptoms of diplodiosis include ataxia and paralysis; however, toxicity studies conducted have not definitively identified DIP as the metabolite responsible for this mycotoxicosis [78]. This is the first report of DIP detected from grown maize in the EC.

The type of ear rot, distribution and prevalence of the species, and the mycotoxins associated are influenced by environmental conditions [13]. Conditions that favour FER caused by *F. annulatum, F. subglutinans* and *F. verticillioides* include high temperatures and dry conditions, while low temperatures and high precipitation favours GER caused by *F. graminearum* and *F. culmorum* [12,14,40]. In general, the EC experiences moderate-to-warm temperatures (19–27 °C) during the day and cooler temperatures (0–7 °C) at night. However, there are variations in climatic conditions between the districts within the EC. For example, the AN district will experience cooler climatic conditions as it is found at a higher elevation close to a mountainous region [79]. The ORT district however experiences more humid conditions and moderate-to-high rainfall along its sub-tropical coast, but it also has pockets of mountainous areas [80]. Suboptimal environmental conditions can contribute to mycotoxins not being present at the two farms that had ear rots and mycotoxigenic species present. Variations in the climatic conditions not only impact the distribution of the ear rots observed but also the species and the mycotoxins that are associated with them. This should be studied further to gain a better understanding of the influence climatic conditions have.

Ear rot infection and subsequent mycotoxin contamination can be mitigated following the application of good agricultural practices. This study observed a difference in the dominant species (from *F. verticillioides* to *F. temperatum)* and mycotoxins (from FB_1_ to multiple mycotoxins) previously reported to be associated with ear-rot-affected maize in the EC and South Africa [33,44,46,64,65,66,81,82], noting that both *F. verticillioides* and *F. temperatum* are commonly isolated during our ongoing survey from seemingly healthy grains. We hypothesize that a shift in the species causing ear rots is occurring in South Africa. In order to confirm this, however, more extensive sampling over a longer period of time is needed. In addition, this study has also highlighted the growing concern that multiple species and multiple mycotoxins may be associated with EC maize. Due to the production potential of the EC, continuous maize ear rot and mycotoxin monitoring should be conducted over multiple seasons to obtain a better understanding of the ear rots and mycotoxins typical for the area. This will ensure continuous safe food production with the goal to minimize the financial loss directly associated with yield loss, as well as indirectly associated with consumer-related health impacts. Isolates obtained during this study have been accessioned into culture collections and will be used in future studies that will explore the biotic and abiotic factors that trigger mycotoxin production. This will not only contribute to the development of prevention and management strategies that are unique to the EC microclimate but also sustainable food security.

## 4. Materials and Methods

### 4.1. Ear Rot Survey and Sample Collection

Nineteen emerging maize farms across the EC were surveyed for maize ear rots during May 2021, approximately two weeks before harvesting. Farms were located in the Alfred Nzo (AN), Chris Hani (CH), Joe Gqabi (JG) and OR Tambo (ORT) districts. Disease severity was determined using a disease rating scale of one to nine, where one is resistant and nine is very susceptible [83]. Every fifth ear in a planted row was examined for ear rot and scored using the above disease rating scale. The disease incidence of individual rows was calculated as a proportion of disease-rated maize cobs to the number of maize cobs surveyed [84]. This is presented as a percentage value. Total field disease incidence was determined by averaging the disease incidence for individual rows. Whole symptomatic maize ears were removed from the plant and placed in individual paper bags labelled with the respective sample number. Samples were stored at 4 °C until processing.

### 4.2. Mycotoxin Sample Preparation

Ear rot samples were processed at SAGL (ISO/IEC17025:2017 accredited) and mycotoxin levels determined following the validated methodology published by Meyer, Skhosana, Motlanthe, Louw and Rohwer [33]. Maize kernels were removed from the cob and milled with a 1 mm sieve (Retsch ZM 200 mill). Milled samples were mechanically mixed for 90 min. Subsamples (10 g ± 0.05 g) were weighed out and 40 mL extraction solvent, comprising 50:25:25 ultra-pure water (<18.2 MΩ·cm)–methanol (MeOH, for high-performance liquid chromatography (HPLC), >99.9%)–acetonitrile (AcCN, ACS/HPLC grade, Burdick and Jackson) was added. The samples were blended using an overhead stirrer for 1 min and were subsequently transferred to a 50 mL polypropylene centrifuge tube. Centrifuge tubes were shaken on a mechanical shaker in the horizontal position for 15 min at 260 rpm and subsequently centrifuged for 10 min at 3000 rpm. A volume of 5 mL of the supernatant was aliquoted into a volumetric flask, and 5 mL of the dilution solution (25% MeOH in H_2_O) was added. Sample extracts were filtered into HPLC amber vials using 13 mm, 0.22 µm syringe filters.

### 4.3. Mycotoxin Analysis

Fourteen mycotoxins were monitored for, including AFs, DON, 15-ADON, DIP, FBs, HT-2, OTA, T-2 and ZEN. A standard (10 µg/mL) was prepared using stock solutions of AFs, DON, 15-ADON, FB_1_, OTA, T-2, ZEN, HT-2 (Romer Labs Diagnostic GmbH, Tulln, Austria), FBs (FB_2_ and FB_3_) (Cape Peninsula University of Technology, Cape Town, South Africa) and DIP (University of Pretoria). Maize matrix-matched standards, for quantitative analyses, were prepared from the stock solution using maize samples not contaminated with the analysed mycotoxins. Additionally, negative controls were prepared using only the extraction solvent and the uncontaminated maize sample, and spiked samples (positive controls) were prepared at a concentration of 5 and 50 µg/kg, respectively, in 10 g uncontaminated maize.

Liquid chromatography mass spectrometry (LC-MS/MS) analysis was carried out on an ultra-performance liquid chromatograph (Waters Acquity UPLC, Waters Corp, Milford, MA, USA) with a C_18_ column (Waters Acquity UPLC BEH, 1.7 µm, 50 × 2.1 mm ID) at 30 °C connected to a tandem (triple) quadrupole mass spectrometer (Waters Acquity TQD). The MRM transitions are described in Appendix A. A programmed gradient elution comprising mobile phase A (0.5 mM ammonium acetate (purity ≥ 98%, Sigma-Aldrich/Merck, St. Louis, MO, USA) and 0.1% formic acid (98–100%, Suprapur, Merck)), and mobile phase B (AcCN with 0.1% formic acid) at a column flow rate of 0.4 mL/min from an A:B ratio of 90:10 to an A:B ratio of 10:90 in 15 min was used for the separation of the 14 mycotoxins. The sample injection volume was set at 5 µL. A standard calibration curve with at least six concentrations was constructed (linear, 1/x weighted, origin excluded) for each mycotoxin. The samples were analysed in duplicate, and the mean values were reported. The results were not corrected for percentage recovery. The percentage recoveries, concentration ranges of the calibration curves and LOQs of the 14 mycotoxins are reported in Appendix A.

### 4.4. Fungal Isolation and Identification

Symptomatic maize ears were observed under a stereomicroscope and fungal material transferred using a sterile needle onto 1/4 Potato Dextrose Agar (PDA) supplemented with chloramphenicol (100 mg/L). The plates were incubated for ~7 d at room temperature (±23 °C). Single-spore cultures were prepared following Leslie and Summerell [85], with pure cultures subsequently preserved in 10% glycerol and stored at −80 °C. The cultures were accessioned into the Applied Mycology working collection (CN collection) and representative cultures for each species into the CMW culture collection, both housed at the Forestry and Agricultural Biotechnology Institute (FABI) at the University of Pretoria, Pretoria, South Africa.

Genomic DNA was extracted from 7 d old cultures grown on 1/4 PDA. DNA extractions for *Fusarium* and *Stenocarpella* isolates were carried out using PrepMan^®^ Ultra (Applied Biosystems, Waltham, MA, USA) following the manufacturer’s instructions. For pigmented fungi such as *Cladosporium, Epicoccum* and *Trichoderma*, DNA extractions were carried out using the Quick-DNA^TM^ Fungal/Bacterial Miniprep kit (Zymo Research, Irvine, CA, USA) following the manufacturer’s instructions.

PCR amplifications were made in 25 µL PCR master mixes consisting of 5 µL of BioLine 5 X MyTaq Reaction Buffer (Meridian BioScience, Ohio, OH, USA), 0.50 µL of each primer (10 µM), 17.85 µL of demineralized sterile water and 0.15 µL of BioLine MyTaq DNA Polymerase (Meridian BioScience, Ohio, OH, USA). The internal transcribed spacer (ITS1-5.8S-ITS2) rDNA region (ITS) [86,87], large subunit (LSU) [88,89] and translation elongation factor 1-alpha (*TEF*) [48,90] gene regions were amplified. PCR conditions and primers used for these are included in Appendix A.

Amplified fragments were purified using Exo-SAP-IT PCR Product Cleanup Reagent (ThermoFischer Scientific, Waltham, MA, USA) and sequenced in both directions using the BigDye Terminator v.3.1 Cycle Sequencing Kit (Applied Biosystems, CA, USA) with the same primers used for PCR. Sequencing reactions were run on an ABI PRISM 3100 Genetic Analyzer (Applied Biosystems, CA, USA). Contigs were assembled and edited using Geneious Prime v. 2019.2.1 (BioMatters Ltd., Auckland, New Zealand).

Sequences generated in this study were compared to locally curated databases, mainly compiled from published taxonomic studies, and obtained from the National Centre for Biotechnology Information’s (NCBI) nucleotide database (http://www.ncbi.nlm.nih.gov/nucleotide/, accessed on: 28 January 2024), Fusarium-MLST (https://fusarium.mycobank.org, accessed on: 28 January 2024) and Fusarium-ID v.3.0 [91] databases. Newly generated sequences were submitted to GenBank.

## Figures and Tables

**Figure 1 toxins-16-00095-f001:**
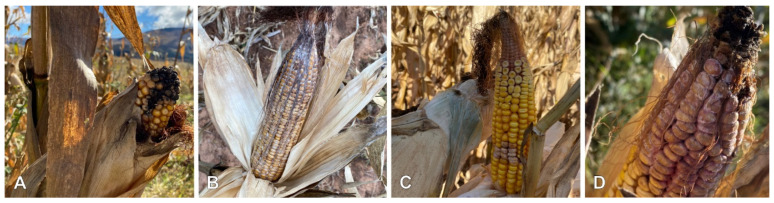
Ear rots collected across the Eastern Cape. (**A**) Cladosporium ear rot, (**B**) Diplodia ear rot, (**C**) Fusarium ear rot and (**D**) Gibberella ear rot.

**Figure 2 toxins-16-00095-f002:**
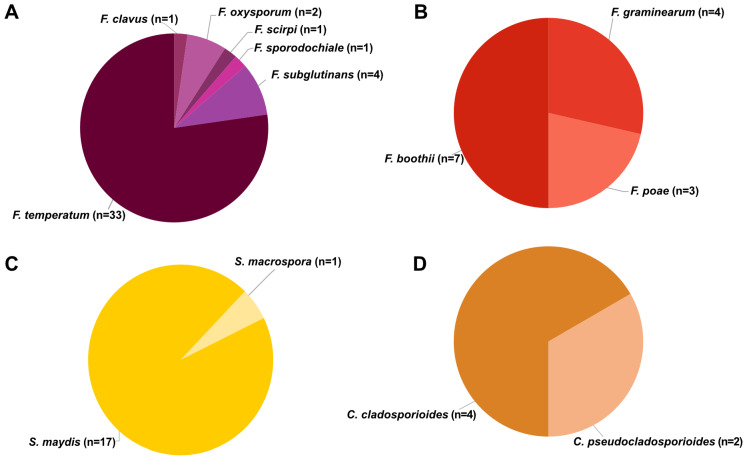
Species isolated from different ear rots in the Eastern Cape (n = number of strains isolated). (**A**) *Fusarium* species isolated from FER, (**B**) *Fusarium* species isolated from GER, (**C**) *Stenocarpella* species isolated from DER and (**D**) *Cladosporium* species isolated from CER.

**Table 1 toxins-16-00095-t001:** Species isolated from ear rots from the respective farms in this study.

Farm Number	Ear Rot	Species Detected	District ^e^	General Disease	
				Incidence (%)	Severity (Average)
Farm 1	CER ^a^	*C. pseudocladosporioides **	AN	14	1
	DER ^b^	*S. maydis ***			
	FER ^c^	*F. temperatum ****			
Farm 2	DER	*S. maydis*	AN	6	1
	FER	*F. temperatum*			
Farm 3	FER	*F. temperatum*	AN	2	2
Farm 4	CER	*C. cladosporioides*, *C. pseudocladosporioides*	CH	3	2
	FER	*F. temperatum*, *F. scirpi*			
Farm 5	DER	*S. maydis*	CH	35	7
	FER	*F. oxysporum*, *F. temperatum*			
	GER ^d^	*F. boothii*, *F. poae*			
Farm 6	DER	*S. maydis*	CH	5	2
	FER	*F. clavus*, *F. oxysporum*, *F. subglutinans*, *F. temperatum*			
Farm 7	FER	*F. oxysporum*, *F. temperatum*	JG	3	2
	GER	*F. boothii*, *F. graminearum*, *F. poae*			
Farm 8	CER	*C. cladosporioides*	JG	6	1
Farm 9	CER	*C. cladosporioides*	JG	11	1
Farm 10	DER	*S. macrospora*, *S. maydis*	ORT	2	2
	FER	*F. temperatum*			
Farm 11	FER	*F. subglutinans*, *F. sporodochiale*	ORT	1	2
Farm 12	CER	*C. cladosporioides*	ORT	0.8	1
	DER	*S. maydis*			
	FER	*F. temperatum*			
	GER	*F. boothii*			

* C. = *Cladosporium*, ** S. = *Stenocarpella* and *** F. = *Fusarium*. ^a^: Cladosporium ear rot (CER); ^b^: Diplodia ear rot (DER); ^c^: Fusarium ear rot (FER) and ^d^: Gibberella ear rot (GER). ^e^: Districts: AN = Alfred Nzo; CH = Chis Hani; JG = Joe Gqabi; and ORT = OR Tambo.

**Table 2 toxins-16-00095-t002:** Mycotoxins detected from ear rots in this study.

Farm Number	Mycotoxins Detected (Average µg/kg)			
	DON ^a^	15-ADON ^b^	DIP ^c^	ZEN ^d^
	LOQ ^e^: 100 µg/kg	LOQ: 100 µg/kg	LOQ: 50 µg/kg	LOQ: 20 µg/kg
	LOD ^f^: 50 µg/kg	LOD: 50 µg/kg	LOD: 25 µg/kg	LOD: 10 µg/kg
Farm 1	ND	ND	726	ND
Farm 2	879	174	122	ND
Farm 3	ND	ND	ND	ND
Farm 4	3143	301	ND	ND
Farm 5	5678	257	33,808	23
Farm 6	254	ND	2648	ND
Farm 7	18,306	1263	ND	639
Farm 8	<LOQ	ND	ND	ND
Farm 9	ND	ND	<LOQ	ND
Farm 10	ND	ND	1268	ND
Farm 11	ND	ND	ND	ND
Farm 12	7273	736	23,004	67

^a^: Deoxynivalenol (DON); ^b^: 15-acetyldeoxynivalenol (15-ADON); ^c^: diplodiatoxin (DIP) and ^d^: zearalenone (ZEN). ^e^ and ^f^: Limit of quantitation (LOQ) means the lowest concentration level that can be quantified with acceptable precision and accuracy by the mass spectrometer. Limit of detection (LOD) is the lowest concentration level that can be detected but not quantified and is 50% of the LOQ of each mycotoxin. A concentration measured below the LOD is reported as not detected (ND).

## Data Availability

Sequence data is available on GenBank. Accession numbers are provided in Appendix A.

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
