# Peer review of "Fungal Species and Mycotoxins Associated with Maize Ear Rots Collected from the Eastern Cape in South Africa"

_toxins, 2024, doi:10.3390/toxins16020095_

Round 1

Reviewer 1 Report

Comments and Suggestions for Authors

General comments:

I have reviewed the manuscript entitled “Fungal species and mycotoxins associated with maize ear rots collected from the Eastern Cape in South Africa” (toxins-2811357) carefully. In this work, the fungi causing maize ear rots were isolated and identified in Eastern Cape in South Africa. Moreover, the contaminations of 14 mycotoxins in commercial maize samples were determined by LC-MS/MS analysis.

Overall, it is not a manuscript submit to Toxins, but for another journal. It looks like a transferred manuscript. Many revisions should be made according to the journal’s template.

My comments are as follows.

Specific comments:

Abstract section:

1.     Line 11, correct “samples respectively” to “samples, respectively”.

2.     Lines 14-15, I suggest the authors to delete the sentence “Fusarium verticillioides were not isolated and fumonisins not detected during this survey”.

Introduction section:

3.     Lines 37-44, these are common knowledge on maize ear rot pathogens, I suggest the authors to delete these sentences.

4.     Lines 27-51, I highly suggest the authors to separate the paragraph into two independent paragraphs, the logic of the current version is chaotic.

5.     Lines 52-54, this sentence is very disharmonious throughout the paragraph. Delete it or put it in other places.

6.     Lines 34-36, Lines 54-56, the two sentences are repetitive.

Materials and Methods section:

7.     Line 79, “nine,  whear…” should be “nine, whear…”.

8.     Line 83, “surveyed(Pierce, 2016).” should be “surveyed (Pierce, 2016).”

9.     Section “2.3. Mycotoxin analysis”, what is the basis for the authors’ choice to analyses the mentioned 14 mycotoxins? Why did not the authos investigate the 3ADON, NIV toxins? As the two are also common Fusairum trichothecene contaminants in cereal products.

10. Line 109, correct “…) respectively” to “…), respectively”.

11. Line 123, correct “Fungal isolations and identification” to “Fungal isolation and identification”.

12. Line 139, correct “5X…” to “5 X   ”.

13. Line 143, the current version of Supplementary Table S1 is very bad, does not make sense. Detailed information should be provided such as primer sequences, PCR conditions, and gene sequenced.

14. The TEF gene accession number for strain CN130C2 is not available in Supplementary Table S2. Please provide it.

15. I request the authors to provide the calibration curves made with known amounts of standard compounds as supplementary materials.

16. Lines 151-154, there may be some problems for the DNA sequence-based identification of fungi, especially for Fusarium species. I strongly suggest the authors to compare their DNA sequences to FUSARIUM-ID v.3.0 (DNA sequence-based identification of Fusarium: A work in progress). For me, I cannot accept the current results without further sequence analysis.

Results section:

17. Table 1 must be re-prepared.

18. Table 1, I suggest the authors to identify the fungi isolated from Farm 8 to species.

19. Line 189, the sentence is not accurate; there may be some other mycotoxins in these samples that were not detected in this study, such as fusarium acid, MON, etc.

20. Line 207, “Fungal identifications” should be “Fungal identification”.

21. Line 228, correct “was” to “were”.

Discussion section:

22. Line 273, “-Gibberella” should be “Gibberella”.

23. Lines 295-297, please re-write the sentence.

24. Lines 322-334, As mentioned by the authors in Lines 34-35 that “The distribution of ear rots is dependent on climatic and geographical conditions”, therefore what’s the relationship between their survey results (such as fungal species or compositions) and local climatic? I suggest the authors to mention this in discussion section.

25. There are too many very old references cited in the manuscript.

26. I strongly suggest the authors to re-prepare the manuscript according to the Instructions for Authors of Toxins (https://www.mdpi.com/journal/toxins/instructions).

Comments on the Quality of English Language

Moderate editing of English language required.

Author Response

Thank you for your comments regarding our manuscript, entitled “Fungal species and mycotoxins associated with maize ear rots collected from the Eastern Cape in South Africa”. We believe that the suggestions made by the reviewers have improved the scientific quality and understandability of our manuscript.

In this revision, we incorporated most of the suggestions made by the reviewers, which has resulted in several minor changes to our manuscript, indicated by the track changes.

Our response to the comments and suggestions made by the reviewers can be found in the document titled attachment.

Reviewer 2 Report

Comments and Suggestions for Authors

This manuscript describes a fungal pathogen and mycotoxin survey study conducted in Eastern Cape region of South Africa. I believe this study is important in bringing information from Eastern Cape region to the rest of South Africa and world.

While the manuscript has adequate information and data, there are some shortcomings which need to be addressed to improve the readability and understandability of the manuscript;

Here are the comments that need addressing:

Line 4: focused in (not on);

Line 12, 138, 292: avoid starting sentence with an abbreviation (LC-MS/MS); PCR; DON, and many of places;

Lines 12-14: Add percent of samples with atleast one mycotoxins. The value 33% with a single mycotoxin is adding information, however, percent of samples with atleast (expect it to be >50% based on the information) would be more informative.

Line 43: Revise FSAMC- it should read as FSAMSC;

Line 45: animals or humans – and instead of or

Line 48: list some of the biotic and abiotic factors here;

Line 108-109: revise sentence – there is no need of respectively as source locations are placed beside the mycotoxin.

Line 115: revise sentence to ‘liquid chromatography tandem mass spectrometry’ or ‘liquid chromatography mass spectrometry’

Line 116-117: separate the manufacturer of mass spectrometry and add in a parenthesis with make, model, and company, location) similar to information presented in line 146.

Line 102-122: include information on LOQ, LOD; what were LOQ and LOD  values? How were they determined. Include the LOQ values for each mycotoxin in table 1.

How were LOQ and LOD values treated for statistical purposes? Include a statistical section in the methods to describe this as well as any other statistical methods used on the data.

Present spike/recoveries data for each mycotoxin; this helps in understanding the absolute quantities of these mycotoxins in the samples. Also whether the data was normalized to recoveries?

Line 122: ‘were’ reported.

Table 1: revise table contents to improve readability: Reduce font sizes of LOD, LOQ etc to fit in the table; Headings can be aligned horizontally instead of vertically. Ear rot and species detected columns need revising to remove numerous horizontal line.

Table 1: revise significant figures in the table – 33808 ug/kg can be presented as 33800 (for example);

Good information to add in the discussion: was 3-ADON tested in any of the samples; 3-ADON with same ,/z as 15-ADON and usually tends to separate at the same retention time as 15-ADON cannot be separated; were there any report of 3-ADON in South Africa; Any information on this even from literature would be helpful to the readers as any reader of this article will get this question about 3-ADON in South Africa.

Line 196-196: how did the DON and 15-ADON concentrations compare? What fraction of the DON?

Suggest moving Figure S1 to main manuscript instead of supplementary.

Discussion: Discussion is rather extensive and many time outside the scope of the results and data presented. While some of the information is important, it could be moved to introduction, rather than in discussion. For example, the authors provide information in discussion on mycotoxins that were not part of this study. While this information is relevant and important which can be moved to introduction, rather than in discussion. Revise the discussion to make it relevant to the findings of this study.

Supplementary information: Due to very limited information in the supplementary tables, I suggest incorporating that information into main text.

Suggest including a geographical map of Easter Cape region (sampling regions) relevant to the other locations of South Africa.

Comments on the Quality of English Language

The manuscript has some sentence structuring and grammatical errors that need to be revised and corrected. It is advisable to write the manuscript in third person instead of in first and second person (use of I or We);

Avoid using abbreviation at the start of the sentences;

Avoid using abbreviations extensively which has been observed in the results and discussion sections;

Fusarium, Gibberella, eta, need not be started with uppercase when in middle of sentences.

Author Response

(The authors gave the same response as above.)

Reviewer 3 Report

Comments and Suggestions for Authors

The manuscript deals with the identification of fungal species and mycotoxins associated with maize ear rots from samples collected in the Eastern Cape region, South Africa. 

Overall, the manuscript is well-organized and well written, with conclusion supported by results. Information reported is of general interest for the mycotoxin-community, especially in consideration of the change in fungal biodiversity following climate change. Mapping fungal species and associated mycotoxins is of great importance to carefully plan future mitigation strategies.

The study represents also the first report of diplodiatoxin in maize, and in association with other mycotoxins.

My only concern is about the lack of identification of fumonisins in maize samples. Although this could be explained based on fungal species distribution and their mycotoxin production under different ecological conditions, the lack of FBs could also be due to poor analytical conditions. It is indeed well-known that fumonisins are quite cumbersome to detect together with other mycotoxins, due to different polarities. More specifically, the samples are extracted in 50:25:25 water:methanol:acetonitrile, then diluted 1:1 in 75:25 water:methanol and finally analysed using ammonium acetate and acetonitrile + 0.1% formic acid as eluents. While such conditions could be fine for trichothecenes, FBs might suffer from bad chromatographic separation.

Since the authors did not provide any information about the analytical performance in terms of recovery nor any detailed gradient for separation, I went back to the cited paper (Meyer et al. 2019). Unfortunately, I did not find information in that manuscript, too. In the present manuscript, as well as in Meyers et al. 2019, the authors just explained that the method was accredited at the National level. Although this is a starting point, it is surely not enough to ensure a high-quality performance.

Therefore, considering that the lack of fumonisins and the shift to DON/ZEN production is a very relevant outcome of the study, I strongly suggest the authors to provide more details about the analytical performance to enhance data confidence, in particular recovery and all chromatographic parameters (i.e. gradient, injection volume, retention time, MRM transitions).

In addition, as minor remarks, Table 1 is poorly edited. I do understand that the layout is quite cumbersome, so I would consider splitting it into 2 tables (one for fungi and one for mycotoxins). In addition, I would consider including Figure S1 in the main text, because it can be of interest to the readers.

Author Response

(The authors gave the same response as above.)

Round 2

Reviewer 1 Report

Comments and Suggestions for Authors

Most corrections have been made, however, some of the responses to reviewer's comments are inconvlusive. For example, the authors' response to the query on section "2.3. Mycotoxin analysis". It is noteworthy that the version of FUSARIUM-ID tool that currently in service is FUSARIUM-ID v.3.0, however, the authors cited a version 20 years ago. Thus, I do not believe the authors truly did this analysis.

Minor revise.

Comments on the Quality of English Language

No.

Author Response

The correction was made according to the reviewer's comment

Reviewer 2 Report

Comments and Suggestions for Authors

The authors have revised the manuscript and included all of reviewers comments. 

Comments on the Quality of English Language

Can be improved further.

Author Response

We express our gratitude to the reviewer for their valuable comments.

Reviewer 3 Report

Comments and Suggestions for Authors

All the suggestion have been incorporated, the manuscript matches now criteria for acceptance

Author Response

(The authors gave the same response as above.)
